# Is Stochastic Gradient Descent Near Optimal?

## Abstract

The success of neural networks over the past decade has established them as effective models for many relevant data generating processes. Statistical theory on neural networks indicates graceful scaling of sample complexity. For example, Jeon & Van Roy (2022) demonstrate that, when data is generated by a ReLU teacher network with $W$ parameters, an *optimal* learner needs only $\tilde{O}(W/\epsilon)$ samples to attain *expected error* $\epsilon$. However, existing computational theory suggests that, even for *single-hidden-layer* teacher networks, to attain small error *for all* such teacher networks, the computation required to achieve this sample complexity is intractable. In this work, we fit single-hidden-layer neural networks to data generated by single-hidden-layer ReLU teacher networks with parameters drawn from a natural distribution. We demonstrate that stochastic gradient descent (SGD) with automated width selection attains small *expected error* with a number of samples and total number of queries both nearly linear in the input dimension and width. This suggests that SGD nearly achieves the information-theoretic sample complexity bounds of Jeon & Van Roy (2022) in a computationally efficient manner. An important difference between our positive empirical results and the negative theoretical results is that the latter address *worst-case* error of deterministic algorithms, while our analysis centers on *expected error* of a stochastic algorithm.

## 1 Introduction

Over the past decade, deep neural networks have produced groundbreaking results. To name a few, they have demonstrated impressive performance on visual classification tasks (He et al., 2016), parsing and synthesizing natural language (Devlin et al., 2018; Brown et al., 2020), and super-human performance in various games (Mnih et al., 2013). These achievements establish neural networks as effective models for many relevant data generating processes.

Statistical theory on neural networks indicate graceful scaling of sample complexity. For example, when the data is generated by a ReLU teacher network with $W$ parameters, Jeon & Van Roy (2022) demonstrate that the sample complexity of an *optimal* learner is $\tilde{O}(W)$.

However, existing computational theory suggests that, even for single-hidden-layer teacher networks, the computation required to achieve this sample complexity is intractable. For example, Goel et al. (2020); Diakonikolas et al. (2020) establish that, for batched stochastic gradient descent with respect to squared or logistic loss to achieve small generalization error *for all* single-hidden-layer teacher networks, the number of samples or number of gradient steps must be superpolynomial in input dimension or network width. Furthermore, current theoretical guarantees for all computationally tractable algorithms proposed for fitting single-hidden-layer teacher networks with parameters drawn from natural distributions only bound sample complexity by high-order polynomial (Janzamin et al., 2015; Ge et al., 2017) or exponential (Zhong et al., 2017; Fu et al., 2020) functions of input dimension or width.

In this work, we aim to reconcile the gap between these negative theoretical results and the apparent practical success of stochastic gradient descent (SGD) in training performant neural networks. To do so, we fit single-hidden-layer neural networks to data generated by single-hidden-layer ReLU teacher networks with parameters drawn from a natural distribution. We demonstrate that SGD with automated width selection attains small expected error with a number of samples and total number of

queries both nearly linear in the input dimension and width. This suggests that SGD nearly achieves the information-theoretic sample complexity bounds established in Jeon & Van Roy (2022); Bartlett et al. (2019) in a computationally efficient manner.

An important difference between our empirical results and the negative theoretical results of Goel et al. (2020); Diakonikolas et al. (2020) is that the latter address worst-case error of deterministic algorithms, while our analysis centers on expected error. The focus on expected error is more in line with the information-theoretic sample complexity bounds of Jeon & Van Roy (2022). Our results suggest that such expected-error analyses may be better-suited for understanding empirical properties of neural network learning.

## 2 RELATED WORK

Our work contributes to the literature on the sample and computational complexity of single-hidden-layer networks. To put our work in context, we review related work in this area, grouped into several categories.

### 2.1 STOCHASTIC QUERY LOWER BOUNDS

Most lower bounds on the sample and computational complexity of single-hidden-layer neural networks have been established through the *stochastic query* framework (Goel et al., 2020; Diakonikolas et al., 2020; Song et al., 2017). A stochastic query algorithm accesses an oracle that returns the expectation of a query function within some tolerance. The literature focuses on query functions that enable gradient descent with respect to common loss functions, with one query per gradient descent step.

Aside from the results of Goel et al. (2020); Diakonikolas et al. (2020), which were discussed in the introduction, Song et al. (2017) show that in a setting where the number of samples is less than the product of the input dimension and the width, exponentially many stochastic queries are required.

### 2.2 SAMPLE COMPLEXITY UPPER BOUNDS

Jeon & Van Roy (2022); Bartlett et al. (2019) study the sample complexity of *optimal* learning from data generated by teacher networks, without addressing algorithms or computational complexity.

Bartlett et al. (2019) establish upper and lower bounds on the VC dimension (see Vapnik & Chervonenkis (1971)) of noiseless neural networks. For piece-wise linear activation functions, their work shows that the VC dimension of a network with $W$ parameters and $L$ layers is upper bounded by $O(WL \log W)$ and that there exist networks with $W$ parameters and $L$ layers with VC dimension lower bounded by $\Omega(WL \log(W/L))$. These bounds on the VC dimension translate to both upper and lower bounds on the sample complexity of any probably approximately correct (PAC) learning algorithm (Valiant, 1984). Results in Hanneke (2016) show that for a PAC algorithm that learns up to within tolerance $\epsilon$ and failure rate at most $\delta$, the sample complexity is $\Theta\left(\frac{1}{\epsilon}(\text{VC} + \log\frac{1}{\delta})\right)$. In our context, this implies $O\left(\frac{1}{\epsilon}(WL \log W + \log\frac{1}{\delta})\right)$ sample complexity *for all* teacher networks with $W$ weights and $L$ layers and $\Omega\left(\frac{1}{\epsilon}(WL \log(W/L) + \log\frac{1}{\delta})\right)$ *for some* of these teacher networks.

Jeon & Van Roy (2022) use information theory to study the number of samples required to learn from a noisy teacher network such that the *expected* error is small. Instead of relying on VC dimension, their bounds scale linearly in the rate-distortion function of the neural network. For networks with ReLU or sign activations, their results imply an $\tilde{O}(W/\epsilon)$ sample complexity bound, where $W$ is the total number of parameters, and $\epsilon$ is the expected error.

For *single-hidden-layer* ReLU teacher networks, both works suggest an upper bound on sample complexity that is linear in the number of parameters, up to logarithmic factors. However, no practical algorithm is given. The VC dimension upper bound implies PAC-learnability, and Jeon & Van Roy studies the *expected* performance of an optimal Bayesian learner. An important difference between these results and the negative stochastic query results is that the latter analyze worst-case performance.

## 2.3 CONCRETE ALGORITHMS

A segment of the literature offers concrete algorithms for learning from single-hidden-layer teacher networks (Zhong et al., 2017; Fu et al., 2020; Janzamin et al., 2015; Ge et al., 2017).

In Zhong et al. (2017), to fit the data generated by a *noiseless* single-hidden-layer ReLU network, the weights are first initialized by a tensor method, which guarantees linear convergence under gradient descent with high probability. However, the sample complexity is exponential in the input dimension and the number of hidden neurons when the weights are i.i.d. Gaussians, as we assume in this paper (see Appendix D for details). Fu et al. (2020) adapts the tensor initialization of Zhong et al. (2017) and provides similar results for cross entropy loss, instead of L2 loss.

Ge et al. (2017) design an alternate objective function $G$ such that using SGD to minimize $G$ can recover the parameters of the single-hidden-layer teacher network with high probability. The sample and computational complexity are high-order polynomials in the input dimension and width.

Janzamin et al. (2015) use tensor factorization, Fourier analysis, and ridge regression to fit the data generated by single-hidden-layer teacher networks with high probability. In the case of Gaussian inputs, the sample and computational complexity are high-order polynomial in input dimension and width. Note that the results in Ge et al. (2017); Janzamin et al. (2015) do not contradict the results of Goel et al. (2020); Diakonikolas et al. (2020), since the former construct algorithms that work *with high probability*.

Results from a couple papers that focus on networks with multiple hidden layers bear additional implications if specialized to single-hidden-layer networks. Arora et al. (2014) propose an algorithm that learns a distribution generated by a sparse neural network with sign activation units and random edge weights. When specialized to a single hidden layer this gives rise to an $\tilde{O}(M^3)$ sample complexity bound, where $M$ is the width. Zhang et al. (2016) propose an algorithm for which sample complexity depends exponentially on maximum among neurons of L1 norms of incoming weights for particular activation units.

## 3 PRELIMINARIES

In this section we give necessary definitions for our experiments, much of which is directly adapted from Jeon & Van Roy (2022).

### 3.1 TEACHER NETWORK

We assume that the training algorithm is given a set $S$ of $N$ i.i.d samples

$$S = \{(x_1, y_1), ..., (x_N, y_N)\} \subset \mathbb{R}^d \times \mathbb{R}.$$

The input $X \sim \mathcal{N}(0, I_d)$, and the output $Y$ is produced by a *random* single-hidden-layer teacher network $g$ with noise $W$:

$$Y = g(X) + W : \mathbb{R}^d \to \mathbb{R}.$$

The *random* single-hidden-layer teacher network $g$ is parametrized by $(a, b, \theta)$:

$$g(X) = \sum_{i=1}^{M} \theta_i \text{relu}(a_i^T X + b_i),$$

where $M$ is the width of the hidden layer and $\text{relu}(x) = \max(0, x)$. For the learnable parameters, we assume that for all $i \in [M], a_i \overset{iid}{\sim} \mathcal{N}(0, \frac{1}{d+1} I_d)$, $b_i \overset{iid}{\sim} \mathcal{N}(0, \frac{1}{d+1})$, and $\theta_i \overset{iid}{\sim} \mathcal{N}(0, \frac{1}{M})$. The choice of variances keeps the variance of $g(x)$ relatively fixed across different $d$ and $M$. We further assume that the noise $W \sim \mathcal{N}(0, \sigma^2)$. We denote the hyperparameters for this teacher network by $\gamma := (d, M, \sigma)$. Note that this is a special case of the ReLU data generating process from Jeon & Van Roy (2022).

### 3.2 ERROR

We define test error as $\mathbf{d}_{KL}(P_Y \| P_{\hat{Y}})$, the KL-divergence from the predictive distribution of $Y$ ($P_{\hat{Y}}$) to its true distribution $P_Y$. We assume that the predictive distribution of $Y$ is Gaussian with the same

variance as the real distribution of $Y$, i.e., $\hat{Y} \sim \mathcal{N}(\hat{g}_S(X), \sigma^2)$, where $\hat{g}_S$ is the model trained on $S$. Then, the KL-divergence simplifies to L2 error with respect to the *noiseless* teacher network scaled inversely by the noise (see section 2.6 of Jeon & Van Roy (2022)):

$$\mathbf{d}_{KL}(P_Y \| P_{\hat{Y}}) = \frac{\mathbb{E}\left[(\hat{g}_S(X) - g(X))^2 | g, S\right]}{2\sigma^2}. \tag{1}$$

### 3.3 SAMPLE COMPLEXITY

Our definition of sample complexity is adapted from Definition 4. in Jeon & Van Roy (2022). For any $\epsilon > 0$, we defined the sample complexity $N_\epsilon$ of a training procedure as the minimal number of samples $N$ such that after training on $N$ samples, the *expected* error is at most $\epsilon$:

$$N_\epsilon = \min\left\{ N : \frac{\mathbb{E}\left[(\hat{g}_S(X) - g(X))^2\right]}{2\sigma^2} \leq \epsilon \right\},$$

where $S$ is an iid set of $N$ training samples, and $\hat{g}_S$ is the model trained on this set. Here the expectation is taken over both $X$ and $g$; so this definition of sample complexity captures the *expected* performance of a training algorithm.

### 3.4 COMPUTATIONAL COMPLEXITY

We use the total number of queries to the training data points as a proxy for computational complexity, which we denote by $T$. More concretely, if the algorithm is trained on $m$ batches of size $n$, then the number of queries to the training data points would be $nm$. When each data point is queried, it generates a forward pass and a backward pass. So the actual computation complexity of the algorithm is a product of $T$ and a scaling factor that depends on the fitting model size.

## 4 EXPERIMENT SETUP

In this section we describe how the experiments are conducted. We first describe the experiment pipeline and then discuss the various components.

### 4.1 EXPERIMENT PIPELINE

The experiment pipeline is outlined in Algorithm 1, and the corresponding code is available online (Appendix A). The definition of various parameters and the respective values chosen for the experiments are summarized in Table 1.

Table 1: Summary of Parameters in Experiment

| Parameter | Descriptions | Values Chosen |
|---|---|---|
| $\gamma = (d, M, \sigma) \in \Gamma$ | Hyperparameters of Teacher Network | |
| $d$ | Input Dimension | $\{1, 2, 4, ..., 2^7 = 128\}$ |
| $M$ | Number of Hidden Neurons | $\{1, 2, 4, ..., 2^7 = 128\}$ |
| $\sigma$ | Standard Deviation of Noise | $\{0.1, 0.2\}$ |
| $\epsilon$ | Target Test Error | 1 to 0.01 for $d, M \leq 2^6 = 64$ 
 1 to 0.1 for $\max(d, M) = 2^7 = 128$ |
| $N \in \mathcal{N}$ | Number of Samples | Successive powers of two to reach all target $\epsilon$ |
| num trials | Number of trials to run for each configuration | At least 32 |

To experimentally verify the the dependence of sample and computational complexity on $d$ and $M$, we generate teacher-networks where $d$ and $M$ are increasing powers of two: $(d, M) \in$

---

**Algorithm 1** Experimental Data Generation Algorithm

---

1: **for** each data generation hyperparameter $\gamma \in \Gamma$ **do**
2:      **for** each sample number $N \in \mathcal{N}$ **do**
3:          **for** $i \in [\text{num trials}]$ **do**
4:             $g \leftarrow \text{sample\_g}(\gamma)$
5:             Sample $N$ i.i.d. $(x_1, x_2, ..., x_N)$ according to $N(0, I_d)$
6:             $\forall j \in [N]$, calculate $y_{j \text{ noiseless}} \leftarrow g(x_j)$
7:             $\forall j \in [N]$, calculate $y_j \leftarrow y_{j \text{ noiseless}} + w_j$, where $w_j \overset{iid}{\sim} N(0, \sigma^2)$.
8:             Set $S \leftarrow \{(x_j, y_j) | j \in [N]\}$
9:             $\hat{g}_S \leftarrow \text{train}(\gamma, S)$, logging the number of queries to data points $T_{\gamma,N,i}$.
10:             Evaluate error according to equation 1:

$$\text{error}_{\gamma,N,i} \leftarrow \frac{\mathbb{E}\left[(\hat{g}_S(X) - g(X))^2 | g, S\right]}{2\sigma^2}$$

11:          **end for**
12:          Average over experiments: let

$$\text{error}_{\gamma,N} \leftarrow \frac{1}{\text{num trials}} \sum_{i \in [\text{num trials}]} \text{error}_{\gamma,N,i}$$

     and

$$T_{\gamma,N} \leftarrow \frac{1}{\text{num trials}} \sum_{i \in [\text{num trials}]} T_{\gamma,N,i}.$$

13:      **end for**
14:      Calculate $N_{\gamma,\epsilon}$:

$$N_{\gamma,\epsilon} \leftarrow \min\{N \in \mathcal{N} : \text{error}_{\gamma,N} \le \epsilon\}$$

15: **end for**

---

$\{1, 2, 4, ..., 128\}^2$. Then, we estimate the sample complexity $N_\epsilon$ for target error $\epsilon$ spanning two orders of magnitude ($\epsilon \in \{1, 0.1, 0.01\}$) with a training algorithm that automatically tunes the width. We run the training algorithm on samples $S$ of increasing size $N$ until the test error is below the specified $\epsilon$, and set the smallest such $N$ as $N_\epsilon$. By choosing $N$ to double each time, we estimate $N_\epsilon$ within a factor of 2. The above procedure is performed for noise $\sigma = 0.1$ and $\sigma = 0.2$; and for each configuration, at least 32 trials are performed to reduce the noise in gathered data.

## 4.2 TRAINING

We split the samples $S$ into an internal training set $S_t$ and a validation set $S_v$ using a $80/20$ ratio. We train single-hidden-layer neural networks of different widths on $S_t$ using golden-section search, and select the model with the best performance on the validation set $S_v$. Various details are described below.

### 4.2.1 ARCHITECTURE OF FITTING NETWORK

The fitting network is an single-hidden-layer ReLU network, the same as the teacher network, but with different widths (number of hidden neurons). No explicit form of regularization like dropout or weight decay is used.

To find the best width, we perform golden-section search (`scipy.optimize.golden`) on widths ranging from 2 to $32 + 8 \cdot \max(N, \sqrt{dM} + \max(d, M))$. This maximum width is chosen to allow ample over-fitting, considering either the number of provided samples, or the architecture of the teacher network. Golden-section search is performed on the *logarithm (base 2)* of the width, with tolerance set to 0.25. The motivation behind this scheme is to get close to a good width by searching few points. For example, at most 8 steps are needed to search through widths from 2 to 1000 in this scheme (the number of steps is at most $\frac{\ln(\text{initial range/tolerance})}{\ln(\phi)}$). We believe that model

performance should roughly be a unimodal function of width. So golden-section search should find widths near the optimum. The number of queries $T$ is the sum of the number of queries for each searched width.

### 4.2.2 OPTIMIZATION

To train the network, we use Adam (Kingma & Ba, 2015) with respect to L2 loss. Aside from the learning rate, We use the default parameters from the PyTorch implementation ($\beta_1 = 0.9, \beta_2 = 0.999$). As empirical evidence suggests that small batch sizes generalize better (for example, see Keskar et al. (2017)), we set the batch size to $64$ for a balance between model performance and training speed.

To automatically set the initial learning rate, we adapt the method first proposed in Smith (2017). We start with a very small learning rate (1e-8) and exponentially increase it until the model starts to diverge. We adapt three methods implemented in the fastai library[1] to estimate the best learning rate [2], and use their medium as the initial learning rate. The queries to the data points in the phase are included in the calculation of $T$.

During training, we reduce the learning rate by a factor of 10 when the validation loss plateaus using `ReduceLROnPlateau` from PyTorch (mode='min' and patience=12).

We stop training whenever the best validation loss fails to decrease relatively by more than $1\%$ in $24$ epochs, and use the model corresponding to the best validation loss. For each fitting network, there is a hard cap of 1500 epochs of training, which is typically never reached.

## 5 RESULTS AND DISCUSSION

### 5.1 SAMPLE COMPLEXITY

In Figure 1, we plot $\epsilon N_\epsilon$ against $dM$ (left) and $\frac{N_\epsilon}{dM}$ against $\epsilon^{-1}$ (right) for the different choices of noise ($\sigma = 0.1, 0.2$). In these plots, $\epsilon$ is the average test error, and $N_\epsilon$ is the corresponding number of samples provided. Both the horizontal axis and the vertical axis are drawn in log scale, with equal aspect ratio. In all plots, we included a scatter plot of the points, and a reference line of unit slope in the log plot, which corresponds to a linear fit of the data. In the plots on the left, we also plotted lines corresponding to the median, the mean, and the 95 and 5 percentiles. In the plots on the right, we use locally weighted smoothing (Cleveland, 1979) to estimate the trend, and the $95\%$ confidence interval is produced by bootstrap resampling two-thirds of the data.

As we can see in the plots, $\epsilon N_\epsilon$ is almost proportional to $dM$, for a wide range of $d$, $M$, and $\epsilon$. Both Bartlett et al. (2019) and Jeon & Van Roy (2022) predict the theoretical sample complexity to be proportional to $\frac{dM}{\epsilon}$, up to log factors. So our results indicate that SGD on neural networks (with automatic width selection) can achieve the theoretical sample complexity of "optimal" learners in the case of single-hidden-layer teacher network.

We note that while the dependence of $N_\epsilon$ on $dM$ is very close to linear for big $dM$, the dependence of $N_\epsilon$ on $\epsilon^{-1}$ is noticeably worse than linear for very small $\epsilon$. Additional plots of the dependence of $N_\epsilon$ on $d$ and $M$ for fixed $\epsilon$ can be found in Appendix B.

### 5.2 COMPUTATIONAL COMPLEXITY

We plot the number of queries $T$ against the number of samples $N$ in Figure 2. As in the previous plots, both the horizontal and vertical axis are in log scale and have equal aspect ratio. We include a reference line of unit slope in the log plot, which corresponds to a linear fit of the data.

From Figure 2 we can see that the dependence of $T$ on $N$ is slightly less than linear and so $T = O(N)$. In the previous section, we demonstrated that $N_\epsilon$, the number of samples necessary to achieve test error within $\epsilon$ tolerance, appears to be $O(\frac{dM}{\epsilon})$. Therefore, $T_\epsilon$, the total number of

---

[1]https://docs.fast.ai/

[2]steep, where the loss as the steepest descent; minimum, for a learning rate $1/20$ of where the loss is the smallest; and valley, when the loss is in the middle of its longest valley

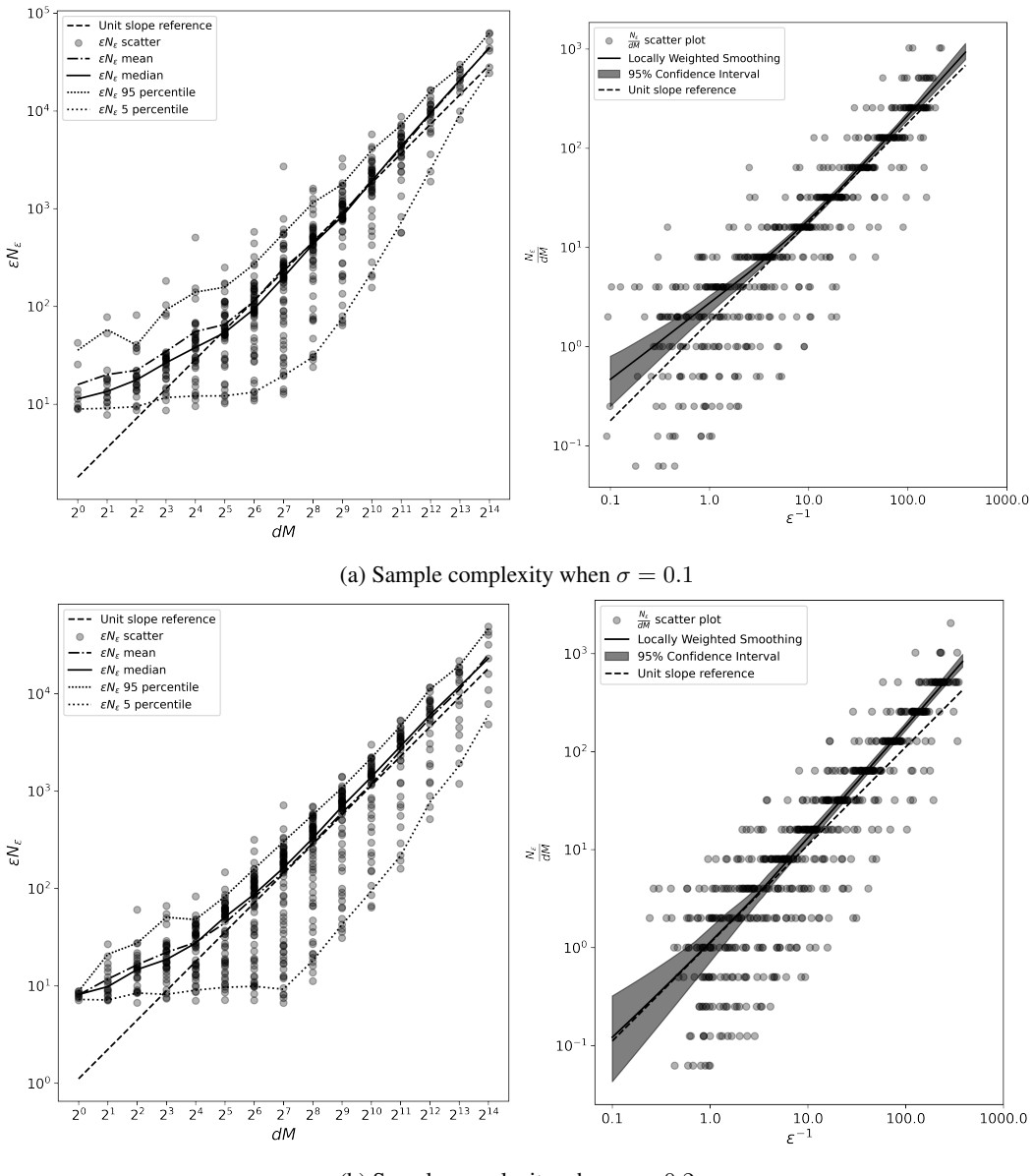

(a) Sample complexity when $\sigma = 0.1$

(b) Sample complexity when $\sigma = 0.2$

Figure 1: Sample complexity is almost linear in $\frac{dM}{\epsilon}$ for wide range of $d$, $M$, and $\epsilon$. $\epsilon$ is the average test error, and $N_\epsilon$ is the corresponding sample size. All vertical and horizontal axises are in the log scale, with equal aspect ratio. A unit slope reference is provided to indicate a linear relationship in the log scale. For $\sigma = 0.1$ (top), the reference lines correspond to $\epsilon N_\epsilon = 1.79 dM$; for $\sigma = 0.2$ (bottom), the reference lines correspond to $\epsilon N_\epsilon = 1.11 dM$. The confidence intervals on the right are generated by bootstrap resampling of two-thirds of the data.

queries to datapoints to achieve $\epsilon$ tolerance, is also approximately proportional to $\frac{dM}{\epsilon}$. This implies that for all $N$, the average number of times *each single* data point is queried is bounded above by a constant.

In our experiments, the width of the fitting network is $O(d + M)$. Since each query of a data point corresponds to at most one forward pass and one backward pass, the overall computational complexity is $O(Nd(d + M)) = \tilde{O}(d^2 M(d + M))$ for fixed $\epsilon$. We hypothesize that by tightening

the upper bound on the fitting network's width to $O(M)$, the current results would still hold, and the corresponding computational complexity could be improved to $\tilde{O}(d^2M^2)$.

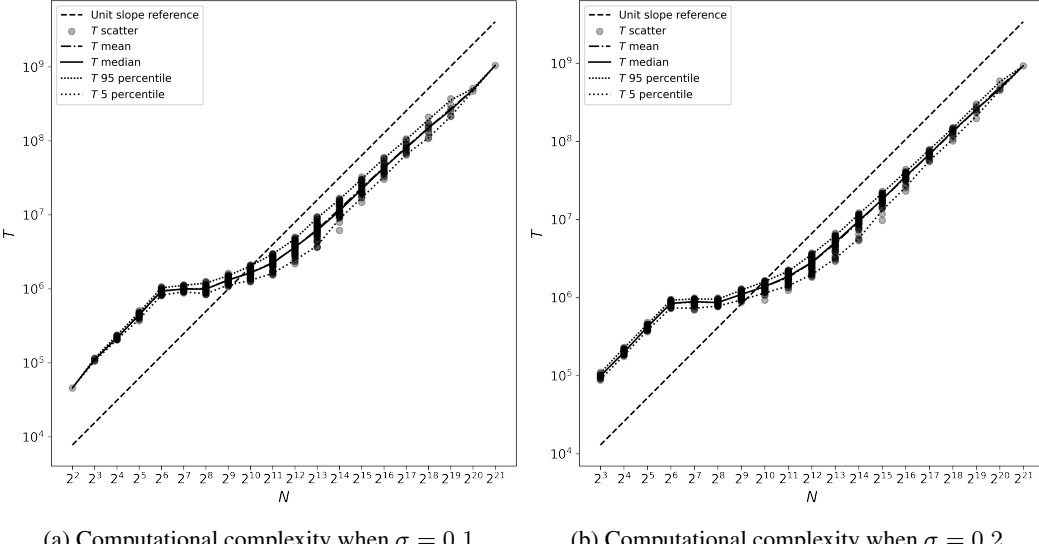

(a) Computational complexity when $\sigma = 0.1$          (b) Computational complexity when $\sigma = 0.2$

Figure 2: Total number of queries to datapoints is sublinear in sample size. All vertical and horizontal axises are in the log scale, with equal aspect ratio. A unit slope reference is provided to indicate a linear relationship in the log scale. For $\sigma = 0.1$ (left), the reference line corresponds to $T = 1940N$; for $\sigma = 0.2$ (right), the reference line corresponds to $T = 1622N$. The sublinear relationship indicates that the average number of times *each single* data point is queried is $O(1)$ for all $N$.

## 6   COMPARISON WITH EXISTING RESULTS

### 6.1   COMPARISON WITH THEORETICAL RESULTS

In the case of single-hidden-layer neural networks, both Bartlett et al. (2019) and Jeon & Van Roy (2022) give theoretical *upper bounds* on the sample complexity that is $\tilde{O}\left(\frac{dN_\epsilon}{\epsilon}\right)$. As for lower bounds on sample complexity, the result in Bartlett et al. (1998) implies the *existence* of single-hidden-layer neural networks[3] with sample complexity at least linear in the total number of weights. However, to the best of our knowledge, there is no tight theoretical lower bound on sample complexity when the teacher network is assumed to be drawn from a distribution, and even for single-hidden-layer teacher networks this seems to remain an open problem.

We empirically demonstrate that for single-hidden-layer teacher networks, running SGD on neural networks with adequate hyper-parameters achieves the best known theoretical bounds on sample complexity, with very manageable run time – the average number of queries per datapoint is constant. SGD empirically works well in terms of sample and computational complexity *in spite of* negative theoretical results in the stochastic query framework (Goel et al., 2020; Diakonikolas et al., 2020). The discrepancy between theory and practice is best explained by the analysis framework. While Goel et al. (2020); Diakonikolas et al. (2020) analyze the *worst case* performance of algorithms and prove that either sample or computational complexity must be super-polynomial, our empirical work studies the *average* performance of SGD. The focus on average case performance is also more in line with the actual uses of neural networks – in practice, people don't necessarily need guarantees that SGD on neural networks works for all datasets, as long as practical algorithms succeed *with high probability*.

---

[3]Mild constraints are imposed on the activation functions. Sigmoid, for example, satisfies the constraints.

## 6.2 COMPARISON WITH OTHER ALGORITHMS

Zhong et al. (2017); Fu et al. (2020); Janzamin et al. (2015); Ge et al. (2017) all construct algorithms to fit single-hidden-layer teacher networks with provable guarantees on sample complexity, computational complexity, and error. Here we highlight some differences between their works and ours:

- While our results indicate sample complexity linear in number of parameters, the mentioned works either have high-order polynomial (Janzamin et al., 2015; Ge et al., 2017) or exponential (Zhong et al. (2017); Fu et al. (2020), see Appendix D for details) sample complexity.
- Our work uses standard machine learning tools (Adam, random weight initialization, early stopping, learning rate decay), while the mentioned works use algorithms not commonly found in practice. Zhong et al. (2017); Fu et al. (2020) uses a tensor method to initialize the weights before applying SGD; Janzamin et al. (2015) use tensor factorization, Fourier analysis, and ridge regression instead of SGD; and Ge et al. (2017) designs an alternate objective function.

## 7 CONCLUSIONS

In this work, we empirically demonstrate that to reach a small *expected* error for single-hidden-layer teacher networks, SGD with automatic width tuning can nearly achieve theoretic sample complexity bounds in a computationally efficient manner. This helps bridge a gap that previously existed between theoretic sample complexity upper bounds and the absence of algorithms that achieve this upper bound computationally efficiently. In addition, the near optimal sample and computation complexity of SGD on neural networks opens up the possibility of modelling it as an *optimal Bayesian learner*. We hope that this new perspective contributes to the general understanding of performance of SGD on neural networks. Investigating whether our results extend to *multiple-*hidden-layer teacher networks remains an interesting question for future research.

**Reproducibility Statement**   We provide the source code for reproducing the experiments (Appendix A).

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

Table 2: Maximum widths for different number of hidden layers.

| Number of hidden layers | Maximum width |
|---|---|
| 1 | $32 + 8 \min\left(T, \sqrt{dM} + \max(d, M)\right)$ |
| 2 | $32 + 2 \min\left(2\sqrt{T}, 2\sqrt{dM} + 2\max(d, M)\right)$ |
| 3 | $16 + 2 \min\left(2\sqrt{T}, 2\sqrt{dM} + 2\max(d, M)\right)$ |

Kai Zhong, Zhao Song, Prateek Jain, Peter L Bartlett, and Inderjit S Dhillon. Recovery guarantees for one-hidden-layer neural networks. In *International conference on machine learning*, pp. 4140–4149. PMLR, 2017.

## A    CODE FOR RUNNING EXPERIMENTS

We anonymously uploaded the source code to https://anonymous.4open.science/r/sample-complexity-4B45/README.md, and will share the git repository upon publication.

## B    ADDITIONAL PLOTS ON SAMPLE COMPLEXITY

For $\sigma = 0.1$ and $\sigma = 0.2$, we plotted the dependence of $N_\epsilon$ on $d$ and $M$ for $\epsilon = 1, 0.1, 0.01$ (see Figure 3 and Figure 4, respectively). Both the horizontal axis and the vertical axis are drawn in log scale, with equal aspect ratio. The dependence on $M$ for different $d$ is shown on the left, and the dependence on $d$ for different $M$ is shown on the right. We use error bars to indicate the range from $\max\left\{N : \frac{\mathbb{E}\left[(\hat{g}_S(X) - g(X))^2\right]}{2\sigma^2} > \epsilon\right\}$ to $\min\left\{N : \frac{\mathbb{E}\left[(\hat{g}_S(X) - g(X))^2\right]}{2\sigma^2} \le \epsilon\right\}$. Since we only run experiments where the sample size $N$ is a power of 2, these two different $N$s always differ by a factor of 2.

From the plots we can see that the dependence of $N_\epsilon$ on $d$ eventually becomes linear (unit slope in our plots) for big $M$. For big $d$, the dependence of $N_\epsilon$ on $M$ eventually becomes slightly worse than linear, but no worse than quadratic (corresponds to slope being 2 in our plots). In addition, these observations hold for $\epsilon$ that spans more than two orders of magnitude.

## C    ARCHITECTURE OF FITTING NETWORK

In this section we study how different fitting network architectures influence performance.

### C.0.1    NUMBER OF LAYERS

We study the performance of the fitting algorithm when the number of hidden layers in the fitting network is 1, 2, and 3. When the number of hidden layers is 2 or 3, we set the number of neurons in each hidden layer to be the same. In all cases, we use golden-section search to find the best width. The minimal width is 2, and the maximum widths are given in Table 2. Again, the maximum width are chosen to allow ample over-fitting, considering either the number of provided samples, or the architecture of the teacher network.

The results are plotted in Figure 5, and summary statistics are shown in Table 3. We see that on average, having only one hidden layer in the fitting network has slightly better performance than having two hidden layers, which in turn has slightly better performance than having three hidden layers. This corresponds well with the idea that the fitting network should have similar architecture as the teacher network.

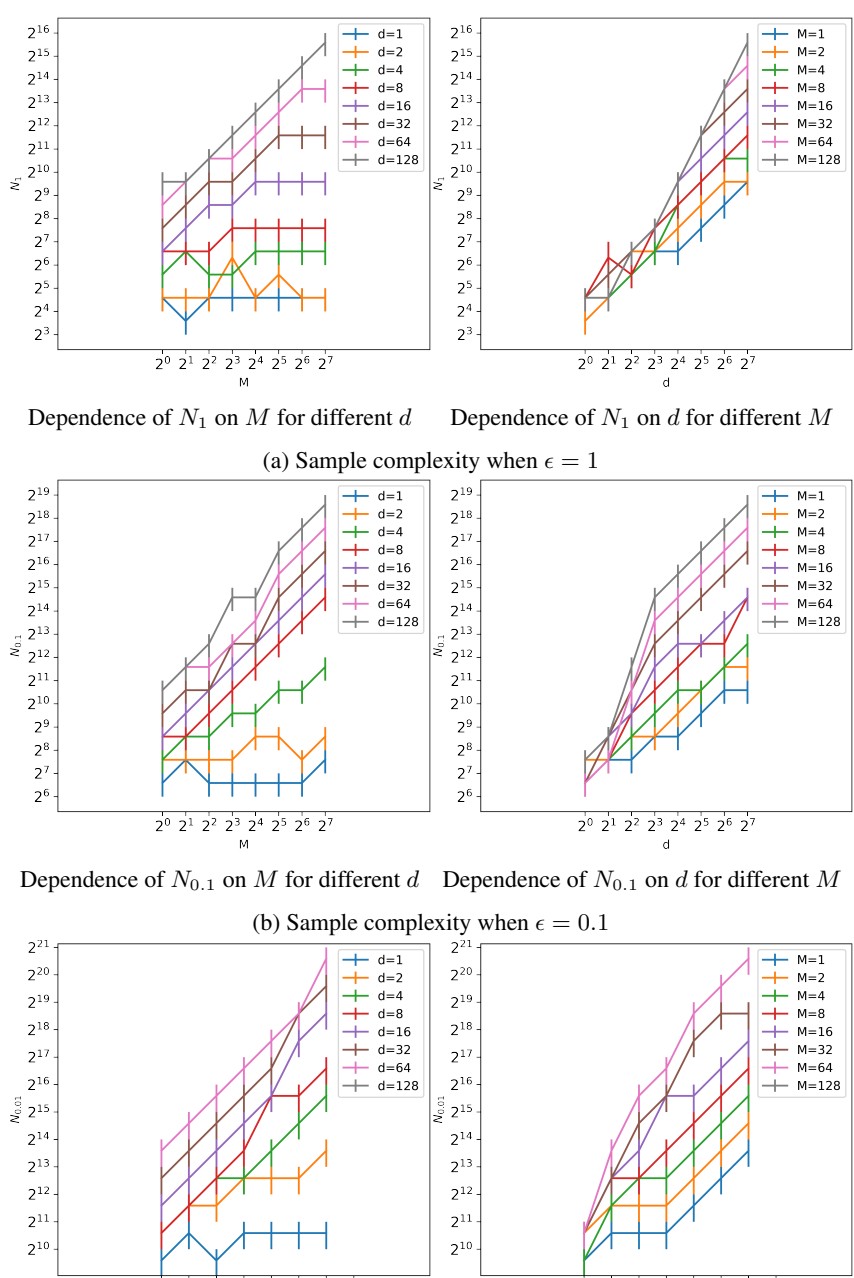

Dependence of $N_1$ on $M$ for different $d$    Dependence of $N_1$ on $d$ for different $M$

(a) Sample complexity when $\epsilon = 1$

Dependence of $N_{0.1}$ on $M$ for different $d$    Dependence of $N_{0.1}$ on $d$ for different $M$

(b) Sample complexity when $\epsilon = 0.1$

Dependence of $N_{0.01}$ on $M$ for different $d$    Dependence of $N_{0.01}$ on $d$ for different $M$

(c) Sample complexity when $\epsilon = 0.01$

Figure 3: Sample complexity $N_\epsilon$ is almost linear in $d$ and $M$ for different $\epsilon$ when $\sigma = 0.1$. All vertical and horizontal axises are in log scale, with equal aspect ratio. The error bars indicate that the estimate of the sample complexity $N_\epsilon$ is within a factor of 2.

### C.0.2 WIDTH OF HIDDEN LAYER

In this part, we fix the fitting network to have only one hidden layer and study the performance of the fitting algorithm for different widths (number of hidden neurons). We use four different schemes to select the width of the fitting network, which we describe in Table 4.

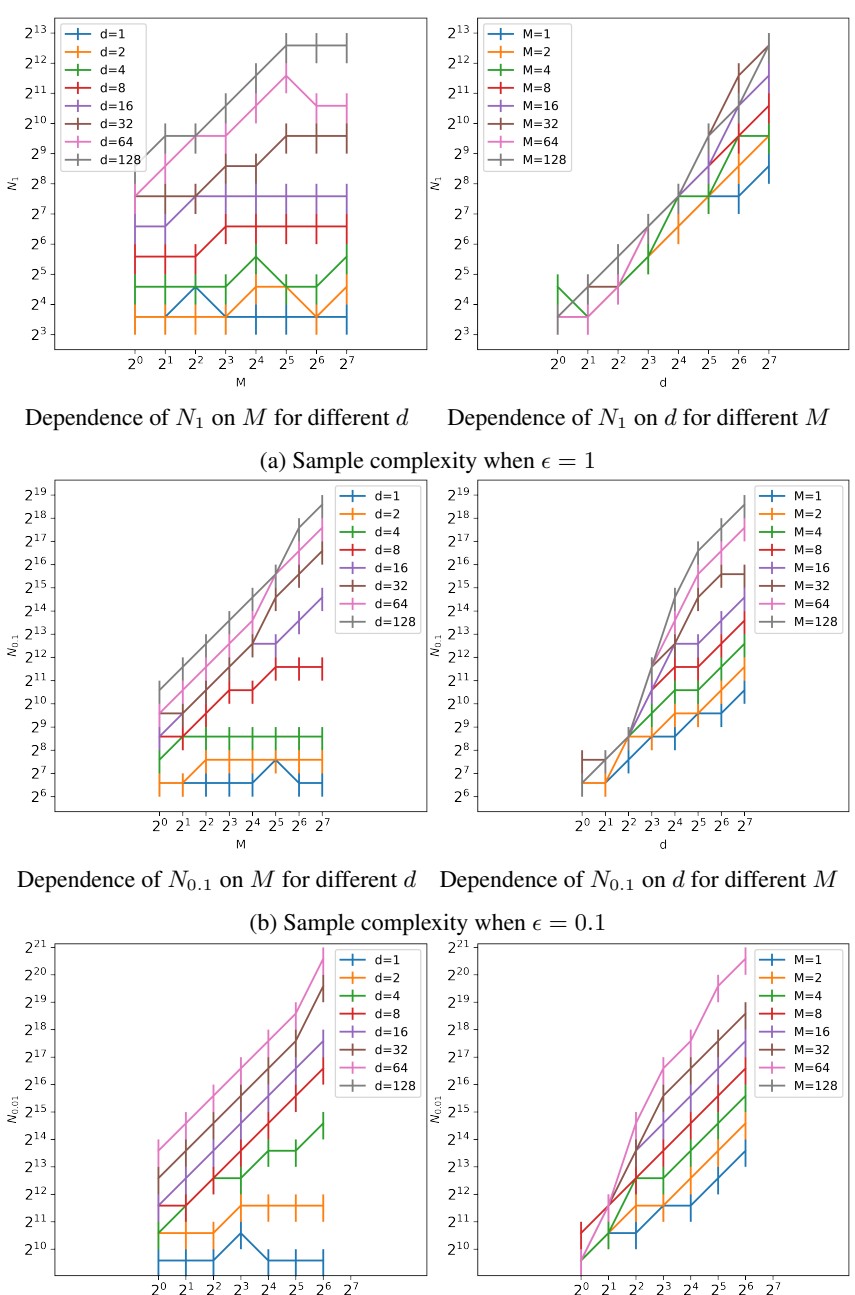

Dependence of $N_1$ on $M$ for different $d$ Dependence of $N_1$ on $d$ for different $M$

(a) Sample complexity when $\epsilon = 1$

Dependence of $N_{0.1}$ on $M$ for different $d$ Dependence of $N_{0.1}$ on $d$ for different $M$

(b) Sample complexity when $\epsilon = 0.1$

Dependence of $N_{0.01}$ on $M$ for different $d$ Dependence of $N_{0.01}$ on $d$ for different $M$

(c) Sample complexity when $\epsilon = 0.01$

Figure 4: Sample complexity $N_\epsilon$ is almost linear in $d$ and $M$ for different $\epsilon$ when $\sigma = 0.2$. All vertical and horizontal axises are in log scale, with equal aspect ratio. The error bars indicate that the estimate of the sample complexity $N_\epsilon$ is within a factor of 2.

We set the width tuning scheme as the baseline, and plot the relative performance of the other schemes in Figure 6, with summary statistics given in Table 5.

The width tuning scheme consistently has the best performance, followed by the **4M** and **best** schemes. The **same** scheme has the worst performance. These results are consistent with empirical observations that over-parametrization is essential in training neural networks (Ge et al., 2017;

Table 3: Geometric mean and median of test error ratio

| Noise | Value | Geometric mean | Median |
|---|---|---|---|
| $\sigma = 0.1$ | $\epsilon_2/\epsilon_1$ | 1.22 | 1.21 |
| | $\epsilon_3/\epsilon_1$ | 1.38 | 1.39 |
| $\sigma = 0.2$ | $\epsilon_2/\epsilon_1$ | 1.14 | 1.16 |
| | $\epsilon_3/\epsilon_1$ | 1.25 | 1.25 |

Table 4: Different schemes of selecting the fitting network width.

| Name | Description |
|---|---|
| **same** | The width of the fitting network is $M$, same as in the teacher network. |
| **4M** | The width of the fitting network is $4M$, corresponding to 4x over-parametrization. |
| **tune** | The width of the fitting network is tuned using golden-section search on the logarithm of the width. The range of widths searched is $$[2, 32 + 8 \cdot \max(N, \sqrt{dM} + \max(d, M))].$$ |
| **best** | Use the median of the widths found by the tune method across trials. |

Table 5: Effect of different width tuning schemes

| Noise | Value | Geometric mean | Median |
|---|---|---|---|
| $\sigma = 0.1$ | $\epsilon_{\text{same}}/\epsilon_{\text{tune}}$ | 2.77 | 1.51 |
| | $\epsilon_{4M}/\epsilon_{\text{tune}}$ | 1.48 | 1.38 |
| | $\epsilon_{\text{best}}/\epsilon_{\text{tune}}$ | 1.59 | 1.21 |
| $\sigma = 0.2$ | $\epsilon_{\text{same}}/\epsilon_{\text{tune}}$ | 1.97 | 1.23 |
| | $\epsilon_{4M}/\epsilon_{\text{tune}}$ | 1.26 | 1.18 |
| | $\epsilon_{\text{best}}/\epsilon_{\text{tune}}$ | 1.28 | 1.10 |

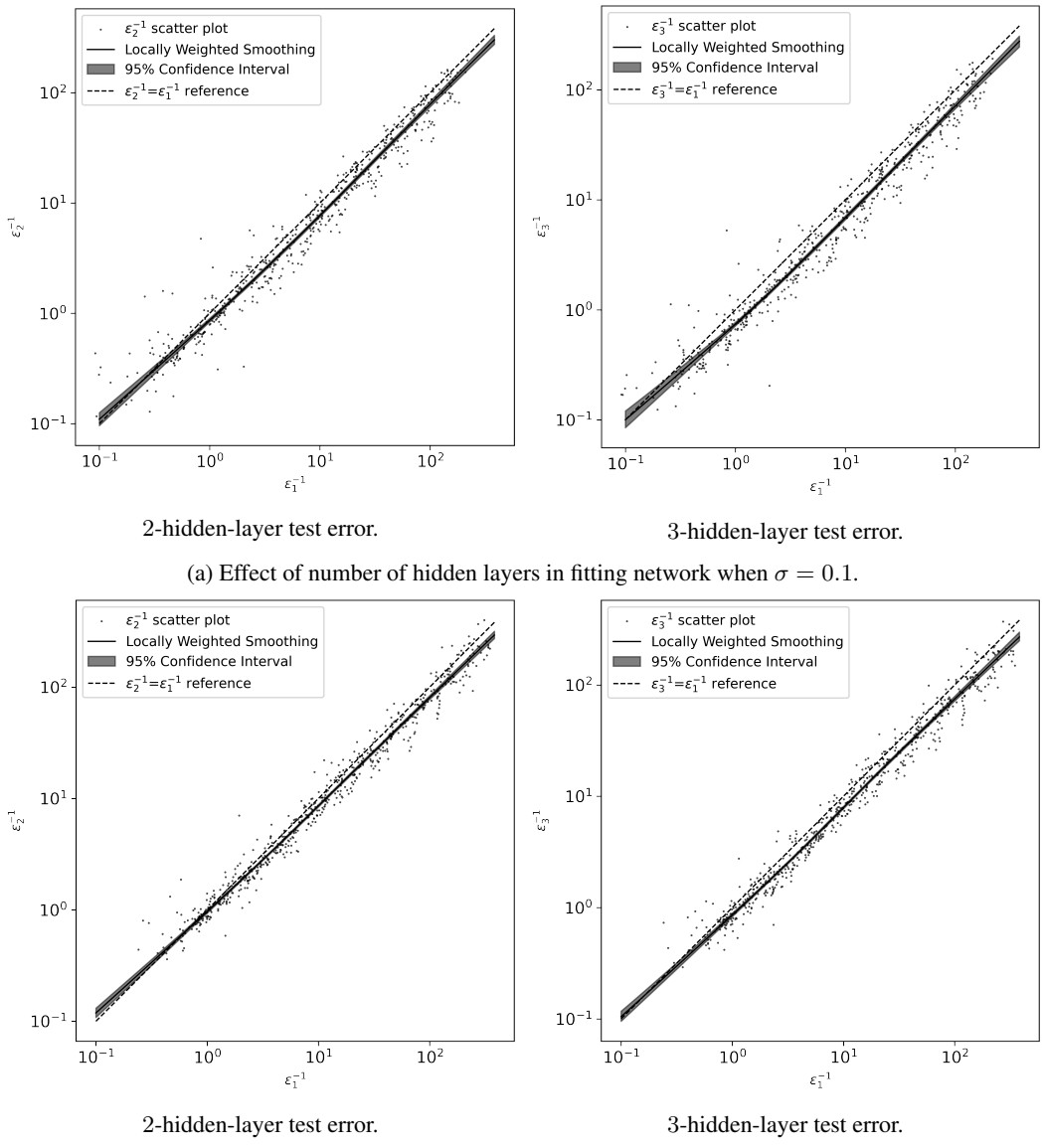

(a) Effect of number of hidden layers in fitting network when $\sigma = 0.1$.

(b) Effect of number of hidden layers in fitting network when $\sigma = 0.2$.

Figure 5: Fitting networks with single hidden layer perform better than those with multiple hidden layers. We plot the inverse of the test error when fitting network has multiple hidden layers ($\epsilon_2^{-1}$ for 2 hidden layers, and $\epsilon_3^{-1}$ for 3) against the inverse of the test error when fitting network has one hidden layer ($\epsilon_1^{-1}$). All axises are in log scale, with equal aspect ratio. A reference line corresponding to equal error is plotted. The region below the line corresponds to single-hidden-layer fitting networks having superior performance. The confidence intervals are generated by bootstrap resampling of two-thirds of the data. In all cases multiple-hidden-layer fitting networks perform slightly worse than single-hidden-layer fitting networks, especially when the test error is small.

Livni et al., 2014; Neyshabur et al., 2018). Perhaps surprisingly, the performance difference between **tune** and **best** also indicates that for optimal performance, the architecture of the fitting networks needs to be tuned to the particular *instantiation* of the teacher network, not just to its architecture and number of samples.

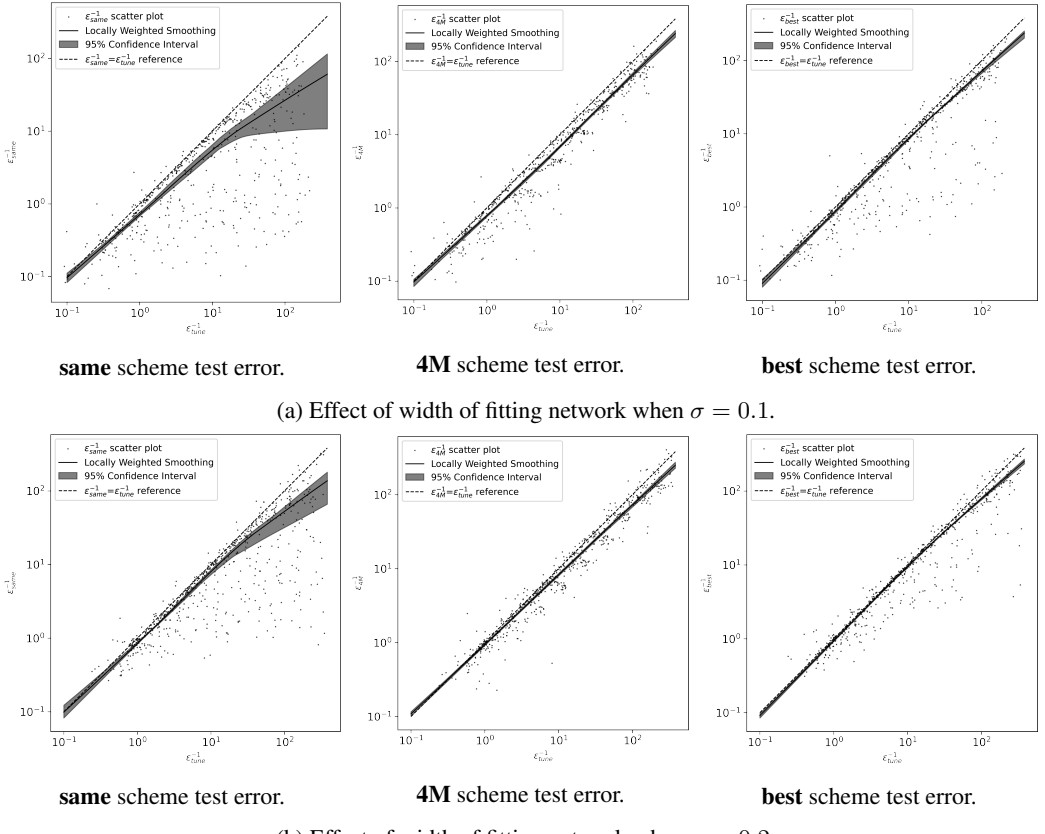

(a) Effect of width of fitting network when $\sigma = 0.1$.

(b) Effect of width of fitting network when $\sigma = 0.2$.

Figure 6: The **tune** scheme has best performance, followed by the **best** and **4M** schemes. The **same** scheme has worst performance. We plot the inverse of the test error when using different schemes to select the width of the fitting network ($\epsilon_{\text{same}}^{-1}, \epsilon_{4M}^{-1}, \epsilon_{\text{best}}^{-1}$) against the inverse of the test error when fitting network automatically tunes its width ($\epsilon_{\text{tune}}^{-1}$). All axises are in log scale, with equal aspect ratio. A reference line corresponding to equal error is plotted. The region below the line corresponds to the width tuning scheme having superior performance. The confidence intervals are generated by bootstrap resampling of two-thirds of the data.

## D    SAMPLE COMPLEXITY IN ZHONG ET AL. (2017); FU ET AL. (2020) FOR IID GAUSSIAN WEIGHTS

The sample and computational complexity of the algorithms in Zhong et al. (2017); Fu et al. (2020) all have a polynomial dependence on a parameter $\lambda$, which depends exponentially on $d$ and $M$ in our setup. [4]

$\lambda$ is defined as $(\prod_{i=1}^{k} \sigma_i)/\sigma_k^k$, where $\sigma_i(W)$ is the $i$-th singular value of the weight matrix $W \in \mathbb{R}^{d \times M}$ of the teacher network. Since Zhong et al. (2017) considers teacher networks where the outer coefficients (our $\theta_i$) are either 1 or $-1$, we need to multiply the outer coefficients inside the activation functions. So with our teacher network, $W$ would be a $d \times M$ Gaussian, with variance $1/d$, multiplied (with broadcasting) with a $1 \times M$ Gaussian, with variance $1/M$.

We set $d = 2M$ and plot $\lambda$ versus $M$ for 1000 trials in Figure 7. The vertical axis is in log scale, and we can clearly see that $\lambda$ depends exponentially on $M$.

---

[4]Zhong et al. (2017) mentions that in the worst case $\lambda$ depends exponentially on the number of hidden units in Remark 4.3.

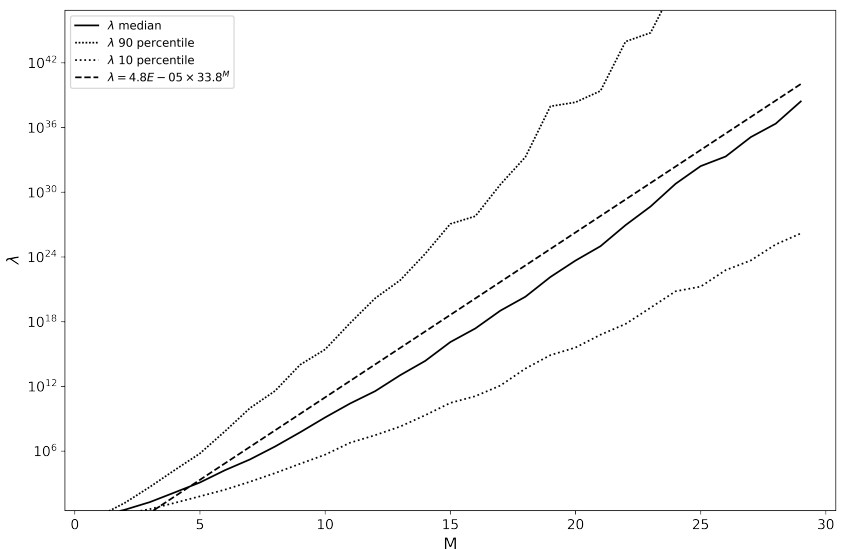

Figure 7: $\lambda$ depends exponentially on $M$ when $d = 2M$.

