# OpenReview forum: "Is Stochastic Gradient Descent Near Optimal?"
_ICLR.cc/2023/Conference — Submitted to ICLR 2023_

### Official Review · Reviewer_nw89 · 2022-10-18

**Confidence:** 3
**Correctness:** 4
**Technical Novelty And Significance:** 2
**Empirical Novelty And Significance:** 3
**Recommendation:** 5

**Clarity, Quality, Novelty And Reproducibility:**

The paper is well-written and easy to follow. However I believe the title can be improved.


**Strength And Weaknesses:**

Strength:
The experimental results are solid with adequate details.

Weaknesses:
1, the title is somewhat ambiguous and over-selling. The scope of experiements is focused on learning single-hidden-layer neural networks, I suggest changing the title to be more specific, for example "SGD learns single-hidden-layer neural networks efficiently".
2, the main result of this paper is empirically showing that SGD learns single-hidden-layer neural networks efficiently. However, this is not surprising because SGD is known to generalize well on large-scale neural networks. Re-showing this phenomenon empirically on toy models doesn't feel interesting to me. It would be interersting to prove it theoretically though.

Question:
Can you give more details on how you generate the parameters of teacher networks? If all the weights are iid Gaussian, is the resulting teacher network still "random"? In particular, in the single-hidden-layer setting, there are works on the infinite width limit of such random neural networks. I'm concerned that the empirical results only applies to the class of fucntions "favored" by random neural networks, which might be very different from practical fucntions we care about.

**Summary Of The Paper:**

This paper empirically demonstrates that SGD learns single-hidden-layer neural networks with near-optimal sample complexity efficiently which matches the theoretical bound of an optimal learner in JV22, while according to theoretically lower bounds achieving the optimal sample complexity is intractable in the worst case.

**Summary Of The Review:**

The empirically results are solid but not significant enough in my opinion, thus I lean to rejection.

---

> ### Author Response · Authors · 2022-11-14
> **Response to Reviewer nw89**
>
> We thank the reviewer for taking the time to evaluate our work. We hope to address some of the concerns in this rebuttal.
>
> The reviewer notes that the title is too broad. We propose a title change to  “Stochastic Gradient Descent is Near Optimal for Single-Hidden-Layer Teacher Networks.”
>
> The reviewer suggests a purely empirical work on single-hidden-layer networks is not interesting. Even though our work is empirical and focuses only on single-hidden-layer teacher networks, we believe that this work is still significant since 1. to the best of our knowledge, no prior work, either theoretical or empirical, has shown that SGD on neural networks can achieve *optimal* sample complexity 2. our work additionally demonstrates that this *optimal* sample complexity to achieve small average error can be achieved computationally efficiently, in spite of theoretical results that give exponential computation bounds when analyzing worst-case error.
>
> The reviewer asks whether the resulting teacher network is still "random” if we choose the weights to be i.i.d. Gaussians. We chose the variances of the weights such that for any fixed input x with L2-norm around sqrt(d), the variance of the output of the noiseless teacher network is roughly constant, independent of the input dimension, and the number of neurons.
>
>  The reviewer also expresses concern that our work only addresses data generated by neural networks, which may be “favored” by neural networks. Our focus on this method follows from a long precedent of learning from data generated from neural networks (e.g., Osband, Ian, et al. "The Neural Testbed: Evaluating Joint Predictions"; Lee, Jaehoon, et al. "Deep neural networks as gaussian processes."). In addition, the authors of the Neural Testbed have demonstrated that results on neural testbeds extend to real datasets. This indicates that testing on data generated by neural networks should be a reasonable way to evaluate algorithms. The NNGP literature shows that infinitely wide neural networks with iid gaussian weight priors are equivalent to a GP prior. When Bayesian inference was performed with respect to this prior on real datasets such as MNIST and CIFAR, they achieved good performance (Lee et al. 2018, Deep Neural Networks as Gaussian Processes). This lends credence to the idea that analyses of data generating processes with iid gaussian weights will have good external validity.

---

### Official Review · Reviewer_CY12 · 2022-10-25

**Confidence:** 2
**Correctness:** 4
**Technical Novelty And Significance:** 2
**Empirical Novelty And Significance:** 2
**Recommendation:** 3

**Clarity, Quality, Novelty And Reproducibility:**

This paper is clear. However, the novelty of this paper may be limited given Jeon &Van Roy (2022).

**Strength And Weaknesses:**

Strength:
1. Well organized, and the message is clear
2. the explanation of synthetic experiments is comprehensive. I believe that the result is reproducible.

Weakness:
1. This paper only conducts experiments for a specific case. In particular, they assume that the teacher network has independent Gaussian weights. Therefore, this is only a special case of the ReLU data-generating process.

2. There are no theoretical results. If one wants to claim that SGD can nearly achieve the information-theoretic sample complexity bounds under a certain condition, one would better derive some rigorous upper bound for the proposed algorithm.

**Summary Of The Paper:**

This paper empirically studies fitting single-hidden-layer neural networks where data is also generated by single-hidden-layer ReLU teacher networks. This paper demonstrates that stochastic gradient descent (SGD) with automated width selection attains small expected errors for a specific experiment setting.   They finally claim that this result further suggests that SGD nearly achieves the information-theoretic sample complexity bounds.

**Summary Of The Review:**

This paper empirically studies fitting single-hidden-layer neural networks where data is also generated by single-hidden-layer ReLU teacher networks. Since there are only some simple experiments for two-layer NN on the synthetic data, this paper would better be viewed as an extended appendix of Jeon &Van Roy (2022). I highly recommend the authors add some theoretical results, even for some special cases.

---

> ### Author Response · Authors · 2022-11-14
> **Response to Reviewer CY12**
>
> We begin by thanking the reviewer for taking the time to evaluate our work. We hope to make a case for our work here.
>
> The reviewer’s first point is that our work regrettably lacks any theoretical results. Despite this, we believe that this work is still significant since 1. to the best of our knowledge, no prior work, either theoretical or empirical, has shown that SGD on neural networks can achieve *optimal* sample complexity 2. our work additionally demonstrates that this *optimal* sample complexity to achieve small average error can be achieved computationally efficiently, in spite of theoretical results that give exponential computation bounds when analyzing worst-case error.
>
> The reviewer notes that we only conduct experiments where the weights of the teacher network are I.I.D. Gaussians. We are curious which other weight distributions the author would be interested in seeing results for. Considering the high dimension of the weights, it would be impractical to experiment over all possible weight distributions.

---

### Official Review · Reviewer_SPUX · 2022-10-26

**Confidence:** 2
**Clarity, Quality, Novelty And Reproducibility:** I believe the work is original. The p…
**Correctness:** 3
**Technical Novelty And Significance:** 2
**Empirical Novelty And Significance:** 2
**Recommendation:** 3

**Strength And Weaknesses:**

strength:

a) The goal of the paper is well-motived.

weaknesses:

a) The title of the paper over claims the purpose of the paper. It only focuses on learning a ReLU teacher network, rather than provides general validation for SGD

b) Even though the current theoretical results only provide worse-case guarantees, I am not fully convinced that the experiments in this paper can illustrate the expected error of an algorithm for the following reasons: i) experiments can only validates some cases and the experiments in the paper are not extensive; ii) there are many versions of SGD, which can hugely affect the results. For example, the paper uses Adam stepsizes, and other stepsize schemes may lead to totally different observationsi.

c)  I also suggest to try networks with more layers.

d) The writing needs improvement. There are same sentences in the abstract and introduction.

**Summary Of The Paper:**

The paper tries to show that SGD is effective in learning a ReLU teacher network. The results seem to suggest that the sample complexity is linear in input dimension and width.

**Summary Of The Review:**

See above.


-----------------------------------------------------------------
I read authors' response and I will keep my score.

---

> ### Author Response · Authors · 2022-11-14
> **Response To Reviewer SPUX**
>
> Firstly, we thank the reviewer for taking time to evaluate our work. They have brought up many excellent points which we hope to address here.
>
> The reviewer notes that the title is too broad. We can understand this sentiment and propose the following new title:  “Stochastic Gradient Descent is Near Optimal for Single-Hidden-Layer Teacher Networks.”
>
> The reviewer suggests experimentation with networks with more layers. We agree that this is an important direction for future research. For our current work, even though our work focuses only on single-hidden-layer teacher networks, we believe that this work is still significant since 1. to the best of our knowledge, no prior work, either theoretical or empirical, has shown that SGD on neural networks can achieve *optimal* sample complexity 2. our work additionally demonstrates that this *optimal* sample complexity to obtain small average error can be achieved *computationally efficiently*, in spite of theoretical results that give exponential computation bounds when analyzing worst-case error.
>
> Lastly, the reviewer notes that the experiments are not extensive enough, and only the Adam optimizer is used. We believe that our experiments are extensive, since 1) the input dimension and width we experimented over all span two orders of magnitude, 2) the target error also spans two orders of magnitude, from 1 to 0.01, and 3) we experimented with noise of different variances. We chose to only use the Adam optimizer, because it is both a commonly used in the deep learning community, and because it was sufficient to demonstrate our point that stochastic gradient descent can achieve optimal rates prescribed by theory.

---

### Official Review · Reviewer_7vtM · 2022-10-28

**Confidence:** 3
**Correctness:** 3
**Technical Novelty And Significance:** 2
**Empirical Novelty And Significance:** 2
**Recommendation:** 3

**Clarity, Quality, Novelty And Reproducibility:**

The paper is written with good clarity. Comments on quality and novelty are summarized in Strength/Weakness. I didn’t check the reproducibility but I’d like to believe the results are reproducible.



**Strength And Weaknesses:**

**Strength**
- It’s of value to empirically measure both the sample complexity and the iteration complexity for learning with neural network of algorithms as simple as SGD. This work further simplifies the dynamic by running simulations on the data generated from a teacher single-hidden-layer Relu network and learning with a student single-hidden-layer Relu network whose width is automatically chosen. The motivation for this setup is straightforward to bring up the gap between the theoretical worst-case guarantees and the empirical performance in expectation.
- The results also showcase the efficiency of SGD in training single-hidden-layer NNs, which may applies to NNs with more advanced structures. The experiments are designed in a well-organized way, and so are the results shown.

**Weakness**
Though many theoretical results were established for single-hidden-layer NNs, and it may be the reason why this work focuses on this simple structure: to directly contrast the results from the theory side and the practice side, results in this work may not bring sufficient insight why SGD trains NNs well  to the deep learning community.


**Summary Of The Paper:**

This work fits the single-hidden-layer neural network to data generated by a teacher network with Gaussian parameters. Conclusion is drawn from experimental results that SGD with automated width selection has both sample and query complexity scaling linearly with the input dimension and width, which complies with the information-theoretic sample complexity bounds and is significantly more computationally efficient than the worst-case query lower bound suggests.


**Summary Of The Review:**

This work focuses on a the open question that how well SGD trains NNs in terms of its sample and query complexity and draws a conclusion matching our expectiaion on empirical performance v.s. theoretical bounds . However, it's heavily limited by the overly simplified NN sturcture and the insufficient insight that could be drawn from the experimets. So I recommand a reject/weak reject.

---

> ### Author Response · Authors · 2022-11-14
> **Response to Reviewer 7vtM**
>
> We thank the reviewer for evaluating our work. We hope to address the issues the reviewer brought up here.
>
> The reviewer believes that the focus on single-hidden-layer networks may not bring sufficient insight to the deep learning community. Even though our work focuses only on single-hidden-layer teacher networks, we believe that this work is still significant since 1. to the best of our knowledge, no prior work, either theoretical or empirical, has shown that SGD on neural networks can achieve *optimal* sample complexity 2. our work additionally demonstrates that this *optimal* sample complexity to obtain small average error can be achieved *computationally efficiently*, in spite of theoretical results that give exponential computation bounds when analyzing worst-case error.

---

### Decision · Program_Chairs · 2023-01-20

**Decision:**

Reject

**Justification For Why Not Higher Score:**

As I have written in "Summary, Strengths And Weaknesses", this paper lacks its novelty. It requires more novel investigation so that this paper would be accepted. The writing is also not satisfactory. The organization of the paper and quality of its presentation can be much improved.

**Justification For Why Not Lower Score:**

N/A

**Metareview: Summary, Strengths And Weaknesses:**

This paper experimentally investigate the statistical/computational efficiency of training neural network by stochastic gradient descent and compare the results by the information theoretic sample complexity lower bound obtained by existing theoretical study (Jeon & Van Roy (2022)).

It is indeed important to investigate the practical efficiency of neural network and validity of some theoretical justification. On the other hand, the main concern raised by the reviewers is that this paper gives no new theoretical work and it is basically rechecking the known results on a toy model. In that sense, this paper lacks its novelty. It is expected that the paper would contain more novel insights so that it could be accepted. Another concern pointed out some reviewers is that the title is overselling because the paper only investigates a very specific two layer neural network. This concern also indicates that the scope of the numerical investigation in this paper is so narrow that it does not give a general insight to the community.